# Anti-Angiogenic Therapy: Current Challenges and Future Perspectives

**DOI:** 10.3390/ijms22073765

**Published:** 2021-04-05

**Authors:** Filipa Lopes-Coelho, Filipa Martins, Sofia A. Pereira, Jacinta Serpa

**Affiliations:** 1Instituto Português de Oncologia de Lisboa Francisco Gentil (IPOLFG), Rua Prof. Lima Basto, 1099-023 Lisboa, Portugal; filipa.coelho@nms.unl.pt (F.L.-C.); filipa.martins@nms.unl.pt (F.M.); 2CEDOC, Chronic Diseases Research Centre, NOVA Medical School, Faculdade de Ciências Médicas, Universidade NOVA de Lisboa, Campo dos Mártires da Pátria, 130, 1169-056 Lisboa, Portugal; sofia.pereira@nms.unl.pt

**Keywords:** neo-angiogenesis, anti-angiogenic therapy, cancer therapy, VEGF, new targets, drug resistance

## Abstract

Anti-angiogenic therapy is an old method to fight cancer that aims to abolish the nutrient and oxygen supply to the tumor cells through the decrease of the vascular network and the avoidance of new blood vessels formation. Most of the anti-angiogenic agents approved for cancer treatment rely on targeting vascular endothelial growth factor (VEGF) actions, as VEGF signaling is considered the main angiogenesis promotor. In addition to the control of angiogenesis, these drugs can potentiate immune therapy as VEGF also exhibits immunosuppressive functions. Despite the mechanistic rational that strongly supports the benefit of drugs to stop cancer progression, they revealed to be insufficient in most cases. We hypothesize that the rehabilitation of old drugs that interfere with mechanisms of angiogenesis related to tumor microenvironment might represent a promising strategy. In this review, we deepened research on the molecular mechanisms underlying anti-angiogenic strategies and their failure and went further into the alternative mechanisms that impact angiogenesis. We concluded that the combinatory targeting of alternative effectors of angiogenic pathways might be a putative solution for anti-angiogenic therapies.

## 1. Introduction

Cancer angiogenesis is a complex process of a new and abnormal blood vessels network formation that accounts for tumor growth and metastasis [1,2]. The angiogenic switch that stimulates endothelial cells (ECs) proliferation and migration to form new blood vessels during tumor growth is promoted by the constant release of pro-angiogenic factors by cancer cells and by cancer-associated stromal cells (e.g., macrophages, fibroblasts, neutrophils, adipocytes) [3,4,5,6,7]. During this process, the increased proliferation of ECs leads to the formation of a disorganized and immature vascular network with disrupted ECs junctions, pericytes detachment and without a continuous basement membrane, responsible for tumor neo-vessels permeability, interstitial fluid pressure, and fragility [1,2,8,9].

While different molecular mediators are important in the control of cancer angiogenesis, the vascular endothelial growth factor (VEGF-A, also known as simply VEGF) is by far the most well studied and targeted in cancer therapy.

### Anti-Angiogenic Therapy, Focused on VEGF, for Cancer Treatment

In 1971, Folkman hypothesized that anti-angiogenic therapy would be beneficial for cancer treatment since it could disrupt the pre-existing blood vessels and avoid the formation of new ones, decreasing oxygen and nutrient supply to cancer cells, and consequently decelerating tumor growth [10,11,12,13]. Decades after Folkman’s statement, antibodies such as bevacizumab, the first VEGF-targeted agent approved by the Food and Drug Administration (FDA) for cancer treatment, was available for cancer therapy [12].

In cancer, neo-angiogenesis is essential for tumor growth and for metastatic processes [11,14,15,16,17]. Despite the existence of other signaling pathways involved in angiogenesis, VEGF/VEGFRs interaction has been considered as a key regulator and constituted an attractive and central target for the development of anti-angiogenic drugs [18,19,20,21,22], the blockade of VEGF signaling pathway by neutralizing antibodies to VEGF or to VEGFRs, soluble VEGFR hybrids, or inhibitors of VEGFRs tyrosine kinase (RTKi) seems to be ineffective as a monotherapy, and resistance is a common event in cancer patients [23]. Therefore, the major challenge in VEGF-targeted therapies is to overcome resistance, due to adaptive and compensatory mechanisms (Figure 1). The limited success of single-targeted monotherapy approaches can be justified by six different mechanisms: (1) The activation of alternative angiogenic signaling pathways; (2) the upregulation of other pro-angiogenic factors [24,25,26,27,28]; (3) the vascular co-option, a process where cancer cells proliferate near the existing blood vessels, avoiding further angiogenesis [29,30]; (4) the vascular mimicry, in which cancer cells acquire an endothelial-like phenotype and led to the formation of blood vessels without ECs involvement [31,32,33]; (5) the endothelial progenitor cells recruitment [34,35], and (6) the increased mobilization of other cell types with a pro-inflammatory/pro-angiogenic phenotype [36]. Supported by this knowledge, a new generation of drugs was developed in order to improve anti-tumoral efficacy, by the simultaneously targeting VEGF signaling pathway and alternative angiogenic pathways. For instance, in vitro and in vivo results showed that the dual targeting of VEGF and fibroblast growth factor (FGF) pathway inhibited ECs proliferation and migration [37]. Moreover, the design and development of new anti-VEGF signaling drugs continues, including the arylamide-5-anilinoquinazoline-8-nitro derivative a recent inhibitor of VEGFR2-kinase activity with in vitro anti-tumor and anti-angiogenic activity [38].

## 2. Targeting Alternative Angiogenic Signaling Pathways to Weaken Cancer

VEGF blockers act on the recognition and neutralization of all bioactive forms of VEGF, preventing VEGFRs activation and consequently inhibiting tumor growth [39]. However, compensatory mechanisms of other angiogenic mediators may be responsible for patients’ resistance to VEGF signaling pathway blockage (Figure 1). For instance, in metastatic colorectal cancer patients, bevacizumab was associated with increased plasma levels of placental growth factor (PIGF), FGF, and platelet-derived growth factor (PDGF), prior or along disease progression [40,41]. Other studies have been reporting the tumor vessels normalization in hepatocellular carcinoma models and the improved efficacy of VEGF-targeted therapy in ocular disease as outcomes of PIGF blockage [42].

Thus, increased levels of other pro-angiogenic factors may compensate VEGF blockage (Figure 1) and trigger alternative VEGF-independent angiogenic pathways.

### 2.1. Fibroblast Growth Factors (FGFs)

FGFs are known as potent angiogenic inducers that increase the proliferation and the migration of ECs [43]. FGF family of tyrosine kinase receptors (FGFR, type V RTK) act upon homo- and heterodimers formation. FGFRs present Ig-like loops in the extracellular domain, which can form covalent dimers through disulfide bonds and promote their constitutive activation [44].

In many cancer types, the anti-VEGF resistant tumors present increased expression of FGF and/or FGFR upon hypoxia [45,46] that contributes for the synergistic cooperation of FGF/FGFR and VEFG/VEGFR axes to the amplification of tumor angiogenesis [25]. In preclinical studies, the application of dual inhibitors of FGF and VEGF pathways have shown an increased anti-cancer efficacy [25,47,48,49,50]. Several inhibitors targeting VEGF and other RTK pathways have been tested, including: Brivanib (dual FGF/VEGF- RTKi), which increases the overall survival (OS) of pancreatic neuroendocrine tumors mouse models; dovitinib (VEGF, FGF and PDGFRs- RTKi) that delays tumor growth; and S49076 (MET, AXL, and FGF RTKi) that induces tumor growth arrest in bevacizumab-resistant tumors [47,49,50]. Nevertheless, in clinical settings after anti-VEGF therapy recurrence, both dovitinib (VEGF, FGF and PDGF RTKi) and nintedanib (VEGF, FGF and PDGF RTKi) were shown to be ineffective [51,52,53]. New applications for old drugs have also been tested in the control of cancer angiogenesis. Thalidomide is a good and promising example, as the teratogenic effect of thalidomide is due to the prevention of the embryo normal angiogenesis through the interference with FGFs [54]. Therefore, the use of thalidomide and analogues as anti-angiogenic drugs has been a focus of cancer research [55,56]. Thalidomide was associated to anti-angiogenic effects in a mice model of neuroblastoma and to decreased tumor growth, metastasis, and angiogenesis in breast cancer [57,58]. Thalidomide also increases the remodeling and stabilization of the abnormal tumor vasculature, in a process mediated by FGF inhibition [59,60,61]. This drug has been also pointed out as a suitable alternative as an anti-angiogenesis effective drug [62], involving the downregulation of FGF and VEGF production [63], in multiple myeloma.

### 2.2. Angiopoietins (ANG)

ANG1 is an angiogenesis suppressor involved in the maturation and stabilization of blood vessels through Tie2 receptor activation (Type XII RTKs) and in the perivascular-ECs interaction and ECs survival [64,65,66]. However, ANG1 also seems to limit the continuous angiogenesis in the tumor and consequently contribute for tumor growth inhibition [67]. ANG1 might be useful in a vascular improvement-based cancer therapy, aiming to the stabilization of the tumor vasculature and further improvement of drug delivery.

On the contrary, ANG2 (mostly produced by ECs) acts as an endogenous antagonist of ANG1 function that leads to remodeling processes or vascular sprouting in response to VEGF [68]. The angiogenic pattern activated by ANG2 resembles the cancer angiogenesis, which is characterized by unstable and leakier vessels [69]. ANG2 is upregulated in many cancer types and has been associated with poor survival and with more invasive phenotypes, which indicates a crucial role in the development of resistance to anti-VEGF therapy [70,71,72]. Colorectal cancer patients with poor responses to bevacizumab had increased ANG2 serum levels, and VEGF and ANG2 blockage in vivo delayed tumor growth, normalized tumor vasculature and increased the survival of mice with glioblastoma [73,74,75]. In pre-clinical studies, the dual VEGF/ANG2 blockage has been shown to suppress revascularization and tumor progression, but their clinical efficacy using vanucizumab (humanized VEGF/ANG2 bi-specific monoclonal antibody) is still under Phase I human trials, though with promising results [75,76,77,78].

### 2.3. Platelet-Derived Growth Factor (PDGF)

In different cancer types aberrant PDGF signaling leads to the secretion of pro-angiogenic factors, promoting an increase in ECs proliferation, migration, sprouting, and tube formation, with a consequent stimulation of lymph-angiogenesis and lymphatic metastasis [77,78,79,80]. PDGF signaling is associated with tumor vascularization in ovarian, non-small-cell lung carcinoma (NSCLC), and hepatocellular carcinoma [79,80]. In glioblastoma, the increased expression of PDGFR is related to poor prognosis [81], and the dual inhibition of VEGF/PDGFR improves survival in mice models [81,82]. In vitro, ponatinib, a multi-tyrosine kinase inhibitor including PDGFR inhibitor, reduces the viability, migration, and tube forming capacity of human umbilical vein ECs (HUVECs) [83]. In the future, the use of PDGF inhibitors might have a place in the portfolio of therapeutic options for the improvement of clinical efficacy of VEGF blockers [84,85].

### 2.4. Hepatocyte Growth Factor (HGF)/c-MET

HGF/c-MET signaling pathway (type X TKR) regulates cell proliferation, motility, and survival, also being a mediator of tumor growth and neo-angiogenesis [86]. In a cancer context, HGF/c-MET activation exerts pro-angiogenic effects, leading to a direct activation of ECs and an indirect stimulation of other pro-angiogenic factors, as VEGF [87]. HGF/c-MET has a role in anti-VEGF therapy resistance and bevacizumab-resistant patients exhibit an upregulation of c-MET expression [88]. In cancer models, it was proven that HGF/c-MET is responsible for the resistance to anti-VEGF therapy with sunitinib (VEGFR and PDGFR RTKi), while the concomitant exposure to HGF/c-MET inhibitors and sunitinib abrogated angiogenesis and tumor growth [89]. In patient-derived glioblastoma cells, VEGF blockage restores MET activity, and c-MET inhibition decreases the invasive capacity of glioblastoma cells [90]. Unfortunately, the concomitant use of onartuzumab (anti-c-MET) with bevacizumab brought no additional clinical benefits [91,92].

### 2.5. Placental Growth Factor (PIGF)

PIGF is a member of the VEGF family that binds to VEGFR1 and its co-receptors neuropilin-1 and 2 (NRP1 and NRP2). PIGF has been indicated as a putative player in anti-VEGF agents resistance since some reports showed its upregulation in patients subjected to anti-VEGF therapy [93,94,95]. Moreover, aflibercept (or ziv-aflibercept), a drug that neutralizes both VEGF and PIGF, showed to be effective in cancer patients-derived xenografts models [96].

### 2.6. Alternative Anti-Angiogenic Factors

In normal conditions, naturally anti-angiogenic factors, such as thrombospondin (TSP-1 and 2), pigment epithelium-derived factor (PEDF), and endostatin, play an important role in the counterbalance of pro-angiogenic factors, through the blockade of multiple pro-angiogenic factors and endothelial cell apoptosis [97]. Given that and considering that the expression of those factors in tumors is often low, there was an increasing interest in the application of these factors and its derivatives as potential anticancer agents [97].

TSP is amongst the first described angiogenesis inhibitors [98], and different attempts were already made to validate TSP and analogs as anti-angiogenic modulators for therapeutic use. For instance, although ATB-510, a TSP mimetic peptide, in a phase I clinical trial showed a favorable low toxicity profile in patients with various solid tumors [99,100], in a posterior phase II trial in soft tissue sarcoma and stage IV melanoma patients ATB-510 did not show efficacy as a monotherapy, since no improvement in patients outcome was achieved [101,102,103]. Likewise, a recombinant version of the three thrombospondin repeat (TSR) domains designated 3TSR was shown to normalize tumor vasculature and to potentiate the tumor regression mediated by chemotherapy [104], reinforcing that a stable vasculature potentiates the anti-cancer therapy by allowing a more effective drug delivery.

PEDF is a multifunctional member of the serine proteinase inhibitor (serpin) family and it plays a role as an inhibitor of angiogenesis [105]. The use of PEGF as a drug has been explored, and different phosphomimetic mutants [106,107] and post-translational modified variants [108,109] with enhanced anti-angiogenic activity were already developed. These strategies have focused mainly on eye disease [110] and wound healing [111], but their application in cancer therapy is under debate [112,113].

Endostatin is a C-terminal fragment of type XVIII collagen and it is the strongest endogenous inhibitor of angiogenesis [114,115], its use in cancer therapy being a promising tool. The use of a recombinant human endostatin (Endostar) was approved by the State Food and Drug Administration of China in 2005 for the treatment of NSCLC patients [116]. Since then, several studies have been relating the anti-cancer properties of this therapy for a variety of other cancer types, including colorectal cancer, melanoma, and breast cancer, whether as a monotherapy or in combination [117,118]. Despite being tested in clinical trials since 2002 [119], further clinical studies are needed to confirm its safety and to better understand endostatin mechanism of action and efficacy, since its effects are complex and may interfere with several different mechanisms [115].

## 3. The Use of Anti-Angiogenic Agents in Cancer: A Disappointing Therapeutic Strategy

The standard care of cancer patients with solid tumors is based on surgical resection followed by chemotherapy and/or radiotherapy, in order to prevent cancer recurrence and the progression of occult microscopic tumors [120,121,122]. However, therapy resistance is still one of the major obstacles for the optimal success of cancer management is [123].

Given the central role of VEGF in the promotion of tumor angiogenesis, its targeting has emerged as the most promising therapeutic strategy for angiogenesis inhibition and cancer treatment. However, decades after Folkman’s statement, anti-angiogenic strategies were developed, and antibodies such as bevacizumab are available for cancer therapy [12]. Nevertheless, so far these strategies have failed, at least in part because the precise molecular mechanisms of cancer neo-angiogenesis remain unclear. Additionally, a novel paradigm emerges since some studies suggested that the abrogation of blood supply will restrict drug delivery (e.g., cytostatic agents) to the tumor, decreasing their clinical efficacy [9]. Additionally, strategies focused on restoring tumor vessels normalization will increase the penetration of therapeutic agents into the tumor, improving the efficacy of drugs [124,125].

VEGF:VEGR axis is considered as the key mediator of pathophysiological angiogenesis. Overactivation of VEGF:VEGFR axis is a trait of many cancer types and correlates with increased microvessel density (MVD) and metastatic spread [18,19,20,22,126]. The blockage of the VEGF:VEGFR axis seems to be ineffective as monotherapy, and primary or de novo resistance is a common feature in cancer patients [23,123]. At the molecular level, the disappointing clinical results obtained by the use of VEGFR inhibitors could be explained by VEGF-independent compensatory mechanisms, though the activation of other angiogenic signaling pathways (e.g., PDGF/PDGFR, FGF/FGFR, Ang/Tie2) (Figure 2) and/or the upregulation of the expression of other pro-angiogenic factors (e.g., bFGF, PDGF) [24,25,26,28,127]. This was clinically evident since cancer patients treated with bevacizumab increase the plasma levels of PIDF, FGF, and PDGF concomitantly with disease progression [40,41]. Accordingly, the in vivo blockage of PIGF or FGF pathway normalizes the tumor vessels and improves the efficacy of VEGF-targeted therapy [25,42,47,48,49,60]. Strategies focused on the dual inhibition of VEGF and other pro-angiogenic signaling pathways might be pivotal for the improvement of anti-angiogenic cancer therapy. For instance, in bevacizumab-resistant tumors, brivanib (dual FGF/VEGF inhibitor) increases the overall survival (OS) in a mouse model of pancreatic neuroendocrine tumor, dovitinib (VEGFR, FGFR and PDGR inhibitor) delays tumor growth, and S49076 (MET, AXL, and FGFR kinase inhibitor) induces tumor growth arrest [47,49,50]. Additionally, in mice models of cancer, the blockage of VEGF/ANG2 suppresses revascularization and tumor progression and increases OS [74,75,76]. Although pre-clinical models showed increased efficacy in tackling tumor angiogenesis, unfortunately the results in clinical settings are not so favorable (Figure 2). The limited success of the anti-antigenic approaches may be also related to the complexity of tumors vascularization processes, as the angiogenic-like phenomena (see Subsection of Section 1) and the recruitment of EPCs (see Subsection of Section 1) [29,30,34,35]. A better understanding on how tumors become vascularized and how they escape from anti-angiogenic therapy will for sure open new perspectives for the development of more effective anti-angiogenic approaches.

## 4. The Versatile Use of Anti-VEGF Agents to Enhance Immunotherapies

VEGF, besides its role as a major driver of angiogenesis, can also have immunosuppressive functions, as the inhibition of immune effector cells and the promotion of immunosuppressive cells. For instance, VEGF through the inhibition of NF-kB signaling pathway is able to inhibit dendritic cell maturation and differentiation [128,129,130], and by the upregulation of PD-L1 (programmed death-protein ligand 1) expression, it inhibits dendritic cell antigen-presentation function, suppressing the further activation and expansion of T-cells [131]. Furthermore, Alfaro et al. demonstrated that VEGF inhibits monocytes differentiation into dendritic cells, being this inhibitory effect reversed upon bevacizumab or sorafenib treatment (VEGFR and PDGFR RTKi) [132].

VEGF-dependent mechanisms promote the generation of an immunosuppressive tumor microenvironment (TME), favoring cancer immune escape and cancer progression [133]. VEGF favors monocyte and macrophage recruitment to the TME [134,135], but it inhibits the differentiation and proliferation of CD8^+^ cytotoxic T-cells [136,137]. VEGFR-2 activation by VEGF on CD8^+^ cells induces the upregulation of immune checkpoint molecules, such as PD-1, TIM-3 (T-cell immunoglobulin mucin receptor 3), and CTLA-4 (cytotoxic T lymphocyte antigen 4), leading to cytotoxic T-cell exhaustion [138]. In contrast, VEGF promotes the proliferation of regulatory T-cells (Tregs), which are involved in tumor development and progression by inhibiting anti-tumor immunity [139,140]. In fact, VEGF levels are positively correlated with the percentage of Tregs, this effect being reversed by VEGF/VEGFR2 blockade in a mouse model of colorectal cancer and by bevacizumab administration in colorectal cancer patients [141,142]. The interaction between VEGF and NRP1 also contributes to the generation of the immunosuppressive microenvironment. Hansen et al. showed that the knockout of NRP1 in Foxp3^+^ Tregs reduced Tregs infiltration into the TME, while it increased CD8^+^ cytotoxic T-cells, leading to prolonged survival in melanoma bearing mice [143,144].

Considering the role of VEGF in the TME, the inhibition of VEGF-induced signaling cascades can ideally suppress tumor growth through two mechanisms: By suppressing angiogenesis and by exerting immunosuppressive effects [20,145]. Due to the dual effect of VEGF on angiogenesis and in the tumor immune microenvironment, several studies addressed the combinatory effect of anti-angiogenic therapies with immune checkpoint blockade.

Cancer immunotherapy is based on the use of immune checkpoint inhibitors (ICIs) directed against immune checkpoint molecules, vital components of immune homeostasis [146]. Cancer cells are able to hijack the expression of these molecules, inducing immune suppression and contributing to tumor evasion of immune surveillance [147,148]. The most common targets of ICIs are the immune checkpoint molecules CTLA-4, PD-1, and its ligands PD-L1 and PD-L2 [149]. Their blockade results in the removal of inhibitory signals of T-cell activation, enhancing anti-tumor immune activity and leading to the inhibition of tumor growth, through CD8^+^ T-cell mediated cancer cells death [150,151]. Although the use of ICIs enhances the immune system of the host towards the recognition and attack of the tumor, the clinical usage of immunotherapy alone only benefit a small subset of patients [152,153,154]. Given the decreased ICIs-cancer patient response and the correlation between anti-angiogenic factors and immunity, anti-angiogenic therapies have been considered as putative enhancers of ICIs treatment.

As previously mentioned, anti-VEGFRs are the first line therapy for patients with metastatic renal cell carcinoma [155]. More recently, their use in combination with ICIs has been tested. Two phase III clinical trials compared the effect of sunitinib versus the combinatory therapy of axitinib (VEGFR and PDGFR RTKi) and pembrolizumab (anti-PD-1 monoclonal antibody) (NCT02853331) [155] and combined with avelumab (anti-PD-L1 monoclonal antibody) (NCT02684006) [156]. In both cases, the combination induced a significantly increase in progression free survival (PFS) of cancer patients [149,157]. Additionally, the combination of axitinib and pembrolizumab ICI prolonged the OS of cancer patients [149]. Interestingly, Wallin et al. observed an increase of intra-tumoral antigen-specific CD8^+^ T-cells migration after the combination with bevacizumab and atezolizumab (anti-PD-L1), in metastatic renal cell carcinoma patients [158].

Two clinical trials (phase I and phase II) are currently testing the combination of ipilimumab (anti-CTLA-4) and bevacizumab in melanoma patients (stage III-IV), focusing on the effect on OS and PFS (NCT01950390) (ClinicalTrials.gov, started in 2013, accessed on 25 March 2021) [159] and on the maximum dose tolerated (NCT00790010) (ClinicalTrials.gov, started in 2008, accessed on 25 March 2021) [160]. In mouse models of melanoma brain metastasis, the combination of axitinib with anti-CTLA-4 antibodies increases the number of effector T-cells and antigen presentation by dendritic cells, which promotes a reduction on tumor growth and an increase in OS [161]. A phase III clinical trial (IMpower150) reported a significant improvement of PFS and OS in metastatic non-squamous subset of NSCLC patients after the combined administration of atezolizumab, bevacizumab, and chemotherapy (carboplatin and paclitaxel) [162]. This result expanded the application of atezolizumab in NSCLC treatment, because its combinatory usage is effective regardless PD-L1 expression [162,163].

Thus far, there is a large panel of promising preclinical and clinical studies ongoing (Table 1). However, a better understanding of the mechanisms involved in both immunotherapy and anti-angiogenic therapy will improve and open new perspectives on the synergistic efficacy, cancer patient safety, and outcome.

## 5. The Potential Use of Other Therapeutic Strategies Targeting Cancer Cells and Impacting Angiogenesis

The clinical results of the use of anti-angiogenic compounds, alone and in combination with conventional cancer therapy, are far from the remarkable successes obtained in pre-clinical settings. Since the efficacy and mechanism of action of the anti-angiogenic drugs (inhibition vs. normalization of tumor vasculature) is still a matter of debate, new efforts have been made for the development of new compounds with both anti-tumor and anti-angiogenic properties. For instance, ivermectin, a FDA-approved anti-parasitic drug for the treatment of intestinal worm infections, inhibit ovarian, breast, and glioma cancer cells growth, and promote cancer cell death [164,165,166,167] while it also targets angiogenesis through the inhibition of capillary network formation, proliferation, and survival of human brain microvascular ECs [167]. In addition, carnosol and carnosic acid, the major components of rosemary extracts, inhibit tumor cell growth and affect several steps of the angiogenesis process, as EC differentiation, proliferation, migration, and proteolytic capability [168]. Other compounds, as fucoidan and sulfated galactofucan have been shown to reduce tumor growth and inhibit tumor angiogenesis, in part through the inhibition of STAT3-regulated genes, as VEGF, Bcl-xL, and cyclin D1 [169,170].

### 5.1. ROS-Related Drugs: A Double-Edge Sword

The metabolic remodeling of tumor and tumor associated stromal cells (TASCs) drive the generation of a pro-oxidant rich-microenvironment, which in turn favors tumor angiogenesis [171,172,173]. The generation of a pro-oxidant and a pro-angiogenic tumor microenvironment seems to work synergistically in the promotion of the angiogenic switch and further tumor angiogenesis [174]. The generation of ROS levels in cancer cells prompted by 27-hydroxycholesterol (27HC) and deferoxamine (DFO) activates STAT-3/VEGF and ERK1/2/HIF1α signaling pathways, promoting tumor angiogenesis and metastasis [175,176]. Moreover, in human colon carcinoma cells, the mutant p53 triggers angiogenesis through the ROS-mediated activation of VEGF and HiF1α [177].

The modulation of the ROS levels in cancer treatment is a double-edged sword. On one hand, the generation of a pro-oxidant microenvironment in early stages of tumor development, and at moderate and non-toxic levels, activates cancer cell survival signaling cascades (e.g., MAPK/ERK1/2, p38, c-Jun N-terminal kinase (JNK), PI3K/Akt) and prompt tumor angiogenesis through the release of pro-angiogenic factors (e.g., VEGF, FGF) and extracellular matrix degrading enzymes (e.g., matrix metalloproteinases- MMPs). On the other hand, the generation of high and toxic ROS levels by chemotherapy and radiotherapy promotes oxidative stress and further cell death and senescence [178,179,180,181,182]. The in vitro exposure of breast cancer cells to the antioxidant resveratrol reduce the ROS accumulation, decreasing paclitaxel-induced cell death [183,184]. The tamoxifen-induced cytotoxicity in MCF-7 breast cancer cells was regulated by the intracellular concentration of the antioxidant vitamin C, decreasing the levels of ROS and lipid peroxides [185].

As mentioned, ROS generation in the TME, depending on the levels, can exert pro-tumorigenic and pro-angiogenic effects or drive oxidative stress-mediated cell death, pointing out that strategies focused on the modulation of ROS levels and oxidative stress to impair tumor progression, angiogenesis, and metastasis need to be carefully monitored. Baicalein, a phenolic flavonoid compound with antioxidant properties, inhibits cancer proliferation and migration, induces cancer cell death, and disrupts the development of tumor vasculature [186,187,188]. OptiBerry, a anthocyanins-rich berry extract with antioxidant properties, shown to inhibit H_2_O_2_ and TNFα-induced VEGF expression by keratinocytes and to diminish the ability of ECs to form hemangioma, suggesting a putative anti-angiogenic, antioxidant, and anti-cancer effect [189]. Other compounds have also shown strong ROS scavenging activities and anti-cancer activity, as the polyphenol from *T. pallida* and white mulberry (*Morus alba*) [190]. However, the effects of this compound on the tumor vasculature have not been evaluated so far. Contradictory information comes from studies suggesting that strategies focused on the generation of ROS and further oxidative stress induce ECs dysfunction, impairing tumor angiogenesis [191,192]. Although most of the studies support the angiogenic potential of ROS as a therapeutic target for anti-cancer and anti-angiogenic therapy, the effects of those strategies on tumor angiogenesis and their mechanisms of action not only in ECs, but also in the crosstalk between ECs:cancer cells, need to be explored. Further research to a better understanding of disease-specific ROS involvement and their potential as anti-tumor strategy will be pivotal for the discovery of new therapeutic targets and further drug development for cancer treatment.

### 5.2. β-Adrenergic Drugs: Repurposing Existing Drugs for Anti-Cancer and Anti-Angiogenic Clinical Purposes

The activation of the β-adrenergic system by catecholamines, epinephrine, and norepinephrine has been related to the tumorigenic processes, as cancer cell proliferation and apoptosis, and also with vascular events, as angiogenesis [193,194]. In ovarian cancer patients under social isolation, the increased levels of noradrenaline are correlated with tumor grade and stage, suggesting the contribution of the β-adrenergic receptors (β-AR) system in cancer progression [195]. Chronic behavioral stress hormones, as adrenaline and/or noradrenaline, promote pancreas, breast, colorectum, and prostate cancer progression [193,196,197,198,199]. More recently, the expression of β2-AR has been proposed as a useful and novel prognostic factor for patients with clear cell renal carcinoma [200]. Besides the biobehavioral factors that contribute to the activation of β-AR system, some cancer cells (e.g., pancreatic, lung, colon) are able to synthesize and release adrenaline [201,202].

β-AR signaling activation via adrenaline and/or noradrenaline is implicated in the promotion of angiogenesis, through the upregulation of pro-angiogenic factors, as VEGF, IL6, IL8, and MMP2 and 9 and by the upregulation of VEGFR2 [203,204,205]. In prostate cancer xenograft models, β-AR signaling promote tumor angiogenesis [193]. The crosstalk between ECs and cancer cells potentiates tumor angiogenesis mediated by β-AR signaling pathway, inducing the EC activation of Jagged1/Notch intercellular signaling and the metabolic shift of ECs from oxidative phosphorylation to aerobic glycolysis, which represents a critical step during the angiogenic switch [203,206]. Interestingly, in a mice model of ovarian cancer, dopamine administration (antagonize the effect of stress hormones) inhibits tumor angiogenesis and stabilizes the already formed tumor vasculature, enhancing cisplatin delivery and efficacy [207]. These results reinforce that the activation of the AR signaling pathway by catecholamines might be a key event in the tumor angiogenesis cascade.

Considering the modulatory role of β-AR signaling in tumorigenesis, β-blockers showed a promising anti-angiogenic and anti-cancer therapeutic value. Preclinical and retrospective studies highlighted the beneficial actions of β-blockers administration in cancer patients, improving the relapse free survival and decreasing the tumor recurrence, cancer-specific mortality, and metastasis [208,209]. Moreover, the efficacy of Propranolol, a first generation non-selective β-AR antagonist (in some tissues inverse agonist), that acts by competing with catecholamines for the binding to β-adrenergic receptors, was shown to be effective in the treatment of hemangioma, the most common infantile benign tumor that involves the accumulation, proliferation, and differentiation of aberrant vascular structures [210,211,212,213]. Propranolol demonstrated anti-cancer and anti-angiogenic pharmacological properties since its administration in breast cancer patients with arterial hypertension significantly reduced the primary tumor development, nodal/metastatic occurrence, and breast cancer-specific mortality [209]; abrogated the VEGF production [214]; inhibited the noradrenaline-induced HIF1α expression in cancer cells [214] and inhibited the catecholamine-induced signaling between macrophages and ECs [215]. In neuroblastoma, β-blockers (carvedilol, nebivolol, and Propranolol) independently of their selectivity, promoted vincristine-induced tumor regression, in part mediated by the inhibition of tumor angiogenesis [216]. Controversially, an ex vivo study using aortic rings showed that β-blockers’ pro-angiogenic or anti-angiogenic activity is independent of their ability to antagonize catecholamine action. For instance, forskolin (β-AR agonist) a direct activator of adenylate cyclase that increased AMPc production through β-signaling activation, decreases VEGF-mediated microvessel sprouting while increases were observed with Propranolol, metoprolol, and bisoprolol (2nd generation, β1-AR–selective antagonists), and while carvedilol (3rd generation, a nonselective β-AR antagonist with additional alfa blocking activity and antioxidant properties) was unable to affect aortic sprouting [217].

Although it has been suggested that β-blockers could be a putative anti-angiogenic drug, some studies have failed to observe this association, remaining unclear as to how β-blockers mechanistically affects and impairs tumor angiogenesis. Moreover, as mentioned, previous studies demonstrated that inhibitory neurotransmitters induce tumor vessel normalization, which might suggest that the inhibition of the β-adrenergic system by β-blockers will promote tumor vessel normalization. This will in turn increase the efficacy of the delivery of anti-cancer drugs. The favorable safety profile, the low cost and immediate clinical availability, together with the putative cancer patient welfares point to the benefits of repositioning an old drug for new clinical purposes.

### 5.3. The Anti-Angiogenic Modulatory Role of Oxidative Stress and DNA Repair Controllers

As previously mentioned (in Section 5.1), a pro-oxidant rich tumor microenvironment is tightly associated with angiogenesis promotion. Considering that high ROS levels activate VEGF and HIF1α signaling pathways, several therapeutic strategies have been developed to target ROS-induced angiogenesis. The generation of free radicals in the cancer context is not a new topic, since the effect of several chemotherapeutic and radiotherapeutic strategies are mediated by ROS production, which in turn affect signaling cascades responsible for cell survival. For instance, the widely used platinum derivatives, such as carboplatin and cisplatin, increase ROS levels, generating nuclear and mitochondrial DNA adducts [218,219]. These adducts are then recognized by DNA damage repair pathways, triggering apoptosis [220]. However, higher ROS are also associated with chemotherapy resistance [221]. Therefore, the combination of these therapies with ROS modulators (either pro- or antioxidants) has been proposed [222], but its translation to clinics has been difficult to apply.

One key signaling molecule that is activated in response to hypoxia is the nuclear factor erythroid 2 like-2 (Nrf2). This transcription factor is the major one responsible for the redox balance, controlling the expression of antioxidant-response genes [223]. More recently, the role of Nrf2 in DNA damage repair has been explored, and some studies have already shown a potential protective effect of Nrf2 in chemotherapy and radiotherapy-induced DNA damage [224]. For instance, in lung cancer cell lines, the Nrf2/glutathione-mediated antioxidant defense pathway plays an important role in conferring resistance to cisplatin, acting as a good biomarker to predict cisplatin sensitivity [225]. Moreover, Nrf2 also has a role in radiotherapy resistance, facilitating the repair of induced DNA damage through the HDR, in a ROS independent manner [226]. Regarding its role in angiogenesis, this transcription factor is activated by hypoxia and triggers several pathways also activated by HIF-1α, through different target genes, such as HO-1, a member of the heme oxygenase family that activates angiogenesis in a VEGF-independent manner [223]. Therefore, targeting Nrf2 or its downstream targets could provide new therapeutic strategies to impair tumor angiogenesis. Several Nrf2 inhibitors have been studied in the context of cancer therapy, as reviewed in Panieri and Saso, 2019 [227], including brusatol, a quassinoid compound extracted from *Brucea javanica* that was shown to deplete Nrf2 protein levels in mouse hepatoma cells [228]. Moreover, brusatol can also inhibit HIF1*α* accumulation under hypoxia in colon carcinoma cells, abrogating its signaling pathway [229]. Even though these are promising results, it is important to note that some controversy remains on the impact of NRF2 pharmacological modulation on tumor growth [230].

In a study developed by Pastukh et al., an oxidative DNA damage and repair mechanism in the VEGF promoter was described, showing a surprising target of ROS generated by hypoxia [231]. ROS were able to cause modifications in the hypoxic responsive elements (HRE) present in the VEGF promoter, thereby increasing VEGF expression through the involvement of the base excision DNA repair (BER) pathway. By chromatin immunoprecipitation analysis, it was found that HIF1 and BER enzymes (8-oxoguanine glycosylase 1, Ogg1, and redox effector factor-1, Ref-1) were bound on HRE, and DNA strand breaks were introduced. Accordingly, inhibition of BER by downregulation of Ogg1 decreased VEGF expression [231].

Moreover, Kaplan et al. showed in an in vivo ovarian cancer model that cediranib, an anti-angiogenic agent (anti-VEGFR), is able to suppress the expression of homology-directed DNA repair (HDR) factors [232]. This effect occurred not only in a hypoxic context, in which hypoxia is able to inhibit HDR by several mechanisms [233], but also in normoxic conditions through the down-regulation of BRCA1/2 and RAD51 genes [232].

One emerging class of tumor therapeutic drugs is poly(ADP-ribose) polymerase (PARP) inhibitors, which target DNA damage repair mechanisms [234]. PARPs belong to BER system and their inhibition promotes the accumulation of single-strand DNA breaks, which will force the occurrence of double-strand DNA breaks. This way PARPs inhibition will account for the accumulation of single-strand and double-strand DNA lesions and consequently it will also affect the homology-directed DNA repair (HDR) system. Therefore, tumors with BRCA 1/2 mutations are more sensitive to this treatment once the mutated proteins are incapable of repairing double-strand DNA breaks (HDR deficit). Initially, this therapy was only applied to breast and ovarian cancer tumors with this alteration, but its efficacy was later showed in subsets of tumors without HDR deficit. Therefore, the combination of PARP inhibitors with chemo- and radiotherapy is currently under phase II trial (ANLOLA, NCT04566952).

Interestingly, PARP inhibition also affects ECs. A PARP inhibitor, GPI, was shown to abrogate EC migration and the formation of tube-like structures, thus impairing angiogenesis, without compromising ECs viability, both in vitro and in vivo [235]. In the last years, some clinical trials were organized to explore the impact of the combination of anti-angiogenic therapies with PARP inhibitors in cancer treatment. For instance, a phase I/II study, investigating the combination of cediranib and olaparib (PARP inhibitor) in women with recurrent ovarian and breast cancer (NCT01116648) has already shown some promising results, improving PFS in women with recurrent platinum-sensitive ovarian cancer [236]. Another ongoing phase I/II trial is evaluating the PARP inhibitor niraparib tolerability and efficacy in combination with bevacizumab vs. niraparib alone in patients with platinum-sensitive epithelial ovarian cancer (AVANOVA, NCT02354131). Some results already available show a promising activity of this combination therapy resulting in improvement in PFS, regardless of the HDR deficiency status or the chemotherapy-free interval [237,238]. More recently, a following phase III trial is comparing the efficacy of niraparib and bevacizumab combination, with the standard of care treatment (NCT03806049).

It is important to notice that ROS-induced DNA damage does not only occur in the nucleus, but also at the mitochondrial level. Mitochondrial damage, resulting in mitochondrial DNA release to cytosol, is able to dysregulate the Hippo-Yap pathway [239]. This pathway is a key inhibitor of angiogenesis [231,232]. A study with an aging rat model showed that mitochondrial DNA-deficient kidney cells failed to upregulate VEGF in response to hypoxia due to mitochondrial dysfunction, which contributes to impaired angiogenesis [240].

Therefore, the relation between ROS-induced DNA damage and angiogenesis has already provided some new potential therapeutic combinations, which needs to be further explored mechanistically and in clinical settings.

## 6. Final Remarks

The failure of anti-VEGF strategies in the control of cancer can be in part related to two major factors. On one hand, the fact that the precise molecular mechanisms of cancer neo-angiogenesis still have secrets. On the other hand, because the abrogation of blood supply will also restrict drug delivery to the tumor, decrease their efficacy, and promote drug resistance [9]. In line, the paradox in the use of anti-angiogenic drugs arises from new findings showing that instead of eradicating blood supply, strategies focused on restoring the tumor vessels normalization would increase the delivery of therapeutic agents to cancer cells, improving the therapeutic efficacy and impairing cancer cells spread [125].

Interestingly, simvastatin, a statin with antioxidant properties, has been showed to reduce hypoxia-induced endothelium leakage and decrease ROS-induced HIF1α and VEGF expression, attenuating VEGF-derived tumor vessel hyperpermeability and improving cisplatin and cyclophosphamide efficacy [241]. In a theoretical scenario, cancer treatment might rely on multi-mechanisms targeting strategies focused on the induction of cancer cells death and in the promotion of tumor vascular regression or stabilization events (Figure 3). At the same time, these strategies will transform the remaining vessels into a more functional vascular network with decreased permeability, promoting drug delivery and impairing metastasis. Cancer cells under a certain threshold have adaptive antioxidant mechanisms controlling oxidative stress, however, above this threshold, ROS accumulation disrupts redox homeostasis and causes severe damage in cancer cells, ultimately leading to cell death [242]. Given these results, strategies to enhance lethal ROS production in cancer cells have a promising anti-cancer effect.


Angiogenesis-based cancer therapeutic strategies must accompany the microenvironmental and metabolic drift, which tumor cells (malignant and stromal) undergo in order to progress. Therefore, by presenting different metabolic patterns and adaptive redox mechanisms, ECs and cancer cells would present a disparate behavior, and whereas oxidative stress activates ECs and stabilizes blood vessels that will be more competent in drugs delivery; cancer cells would be endangered by oxidative stress and by drugs aggression.


## Figures and Tables

**Figure 1 ijms-22-03765-f001:**
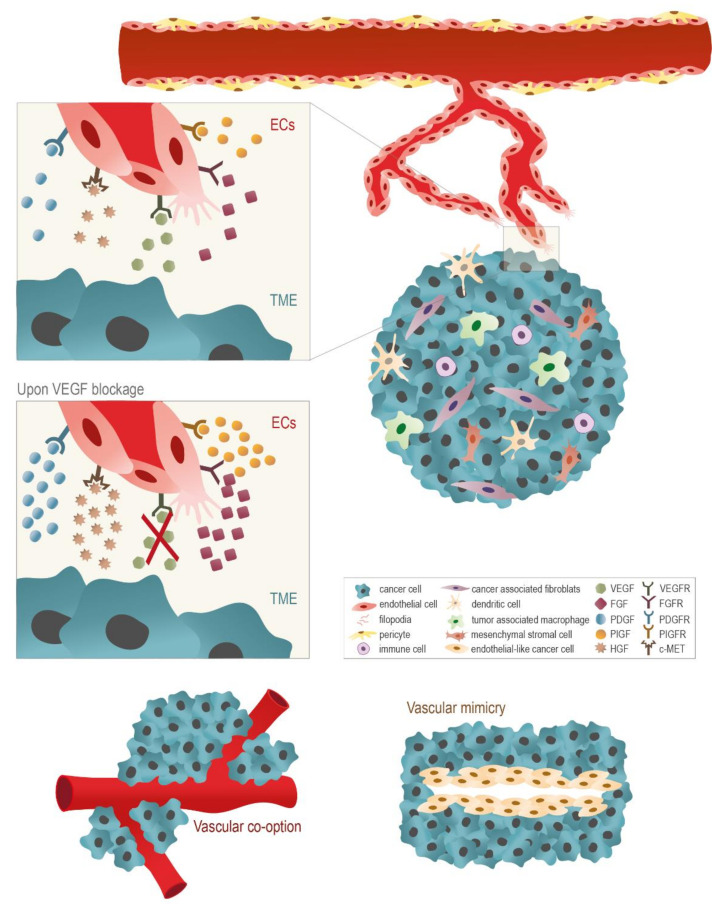
Pro-angiogenic factors released by cancer cells and tumor microenvironment (TME) are essential for neo-angiogenesis promotion, tumor growth, vascular co-option or vascular mimicry. Neo-angiogenesis is stimulated by the release of pro-angiogenic factors by cancer cells or by other cells in the tumor microenvironment (TME), as macrophages, stromal cells, and fibroblast. Among the positive regulators, vascular endothelial growth factor (VEGF) is the most well-studied, and it is almost ubiquitously present at angiogenesis sites. Once it binds to its tyrosine kinases receptors (VEGFRs), they become activated and form homo- or heterodimers, triggering intracellular signaling cascades to stimulate ECs’ proliferation, migration, differentiation, tube formation, and permeability control. Besides VEGF, other pro-angiogenic factors as fibroblast growth factor (FGF), platelet-derived growth factor (PDGF), placental growth factor (PIGF), and hepatocyte growth factor (HGF) are essential for endothelial tip cell activation (endothelial cell (EC)) that respond and guide EC migration in the direction of a pro-angiogenic stimulus, thereby promoting neo-angiogenesis). Moreover, a key process during EC migration is the formation of filopodia (actin-rich cellular protrusions) in tip cells, that will guide ECs towards the pro-angiogenic rich microenvironment. Nowadays, the employment of anti-angiogenic drugs for cancer treatment has been disappointing, in part because most drugs are VEGF-centered. VEGF blockade triggers compensatory mechanisms, such as the upregulation of other pro-angiogenic factors expression and the activation of other angiogenic signaling pathways, leading to patient resistance to VEGF signaling pathway blockade. During vascular co-option, tumor cells migrate along the preexistent blood vessels (specially tumor cells, rich in myofibroblasts), taking over of the existing vasculature to tumor blood supply. This process is essentially reported in highly vascularized tumors, as brain, lung, and liver. Vascular mimicry is characterized by a phenomenon resembling tumor blood supply independently of angiogenesis or endothelial cells (ECs), being correlated with poor patient survival. During this process, cancer cells acquire endothelial-like properties and organize into vascular-like structures acting as a system to obtain nutrients and oxygen, independently of normal blood vessels or neo-angiogenesis. FGF: Fibroblast growth factor; FGFR: Fibroblast growth factor receptor; HGF: Hepatocyte growth factor; PDGF: Platelet-derived growth factor; PDGFR: Platelet-derived growth factor receptor; PIGF: Placental growth factor; PIGFR: Platelet-derived growth factor receptor; VEGF: Vascular endothelial growth factor; VEGFR: Vascular endothelial growth factor receptor.

**Figure 2 ijms-22-03765-f002:**
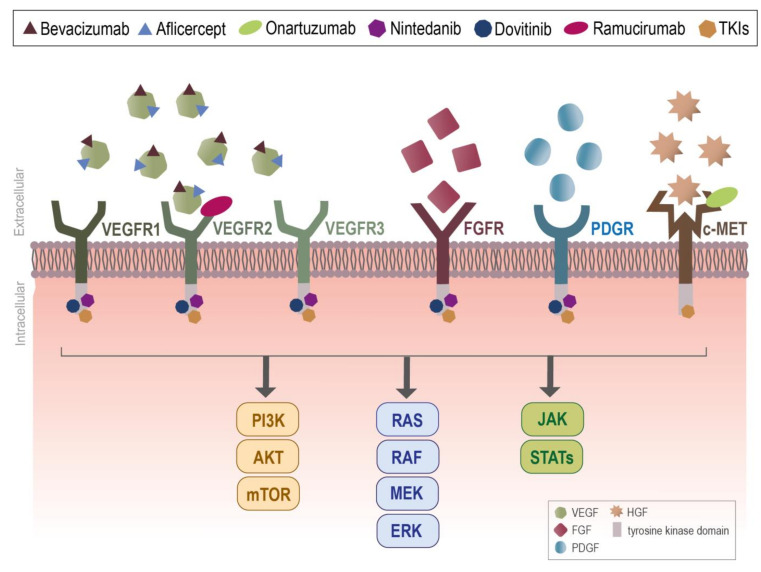
Anti-angiogenic drugs can act as decoy agents, monoclonal antibodies, or tyrosine kinase inhibitors. Bevacizumab was the first anti-angiogenic drug approved for human use, however, its use as monotherapy is inefficient. Compensatory mechanisms of other angiogenic mediators induces a switch in the vascular endothelial growth factor (VEGF)-dependent state to a VEGF-independent angiogenic process and leads to patient resistance to VEGF signaling pathway blockade. Nowadays, a new class of drugs, by blocking other effectors in angiogenic signaling pathways, try to improve the clinical efficacy of anti-angiogenic therapies. In some scenarios, the use of anti-angiogenic therapies delays the disease outcome of cancer patients, still other patients do not have any additional benefits. The major signaling pathways downstream of VEGFRs, FGFRs, PDGRs, and c-MET in endothelial cells (ECs) are PI3K-AKT-mTOR, Ras-Raf-MEK-ERK, and JAK-STATs, which are pivotal for EC survival, proliferation, and migration. AKT: Protein kinase B; ERK: Extracellular-signal-regulated kinase; FGF: Fibroblast growth factor; FGFR: Fibroblast growth factor receptor; HGF: Hepatocyte growth factor; JAK: Janus protein tyrosine kinase; MEK: Mitogen-activated protein kinase; mTOR: Mammalian target of rapamycin; PDGF: Platelet-derived growth factor; PDGFR: Platelet-derived growth factor receptor; PI3K: Phosphoinositide 3-kinases; Raf: Serine/Threonine Kinase; STAT: Signal transducer and activator of transcription protein; VEGF: Vascular endothelial growth factor; VEGFR: Vascular endothelial growth factor receptor.

**Figure 3 ijms-22-03765-f003:**
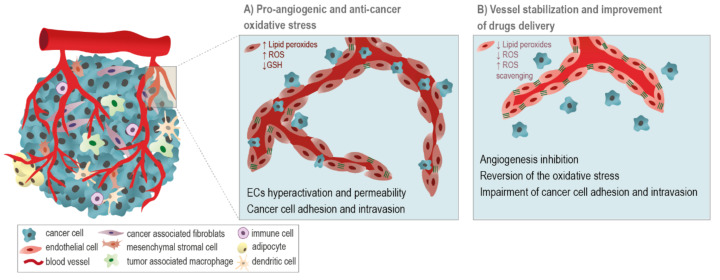
Increased oxidative stress activates endothelial cells (ECs) and kills cancer cells, and antioxidant mechanisms can stabilize vessels and improve anti-cancer chemotherapy. (**A**) A pro-angiogenic oxidative microenvironment, through the increased generation of ROS, the accumulation of lipid peroxides and glutathione (GSH) depletion is implicated in the promotion of ECs hyperactivation, vessels leakiness, and cancer cells migration and intravasation. (**B**) ROS scavenging activity is anti-angiogenic, since on one hand, it impairs the ECs activation, and on other hand, it promotes vessels normalization, pivotal to impair metastasis and improve the delivery of chemotherapeutic agents.

**Table 1 ijms-22-03765-t001:** Ongoing clinical trials evaluating immune-checkpoint inhibitors (ICIs) in combination with anti-angiogenic agents in solid tumors.

Trial Identifier	Treatment (Arm of Combination Therapy)	Comparison	Cancer Types	Study Phase	Primary Endpoint	Registration Date	Status *
**NCT00790010**	ipilimumab + bevacizumab	between cohorts	Melanoma (III-IV)	I	safety, tolerability, max tolerated dose	12 November 2008	Active, not recruiting
**NCT01950390**	ipilimumab + bevacizumab	ipilimumab	Melanoma (III-IV)	II	OS	23 September 2013	Active, not recruiting
**NCT02420821**	atezolizumab + bevacizumab	sunitinib	mRCC	III	PD, PFS, OS	15 April 2015	Active, not recruiting
**NCT02684006**	avelumab + axitinib	sunitinib	mRCC	III	PFS, OS	25 January 2016	Active, not recruiting
**NCT02853331**	pembrolizumab + axitinib	sunitinib	mRCC	III	PFS, OS	29 July 2016	Active, not recruiting
**NCT03914300**	cabozantinib + nivolumab + ipilimumab	-	Thyroid cancer	II	ORR	11 April 2019	Recruiting
**NCT03990571**	axitinib + avelumab	-	Recurrent/metastatic ACC	II	ORR	17 June 2019	Recruiting
**NCT04017455**	atezolizumab + bevacizumab	-	Rectal cancer	II	clinical complete and near-complete response rate	10 July 2019	Recruiting
**NCT04170556**	regorafenib + nivolumab	-	HCC	I/II	Rate of AE	18 November 2019	Recruiting
**NCT04213170**	sintilimab + bevacizumab	-	Brain metastases from NSCLC	II	iPFS, OS, PFS	25 December 2019	Recruiting
**NCT04408118**	atezolizumab + bevacizumab + paclitaxel	-	BC, TNBC	II	PFS	20 May 2020	Recruiting
**NCT04493203**	nivolumab + axitinib	-	Melanoma (III-IV)	II	ORR	22 July 2020	Recruiting
**NCT04727307**	atezolizumab + RFA + bevacizumab + atezolizumab	RFA	Small HCC	II	Recurrence-free survival	22 January 2021	Recruiting
**NCT04732598**	bevacizumab + atezolizumab + paclitaxel	bevacizumab + paclitaxel	BC	III	PFS	28 January 2021	Recruiting

* Status according to https://clinicaltrials.gov/, accessed on 25 March 2021. Immunotherapy drugs: Anti-PD-1: Pembrolizumab, nivolumab, sintilimab; anti. PD-L1: Atezolizumab, avelumab; anti-CTL4: Ipilimumab. Anti-angiogenic therapies: Anti-VEGF: Bevacizumab; anti-VEGFRs: Axitinib, sunitinib, regorafenib, cabozantinib. Chemotherapy: Paclitaxel, carboplatin. ACC: Adenoid cystic carcinoma; AE: Adverse events; BC: Breast cancer; HCC: Hepatocellular carcinoma; iPFS: Intracranial progression free survival; mRCC: Advanced/metastatic renal cell carcinoma; NSCLC: Non-small cell lung cancer; ORR: Overall response rate; OS: Overall survival; PD: Disease progression; PFS: Progression free survival; RFA: Percutaneous radiofrequency ablation; TNBC: Triple negative breast cancer.

## Data Availability

Not applicable.

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
