# Peer review of "Anti-Angiogenic Therapy: Current Challenges and Future Perspectives"

_ijms, 2021, doi:10.3390/ijms22073765_

Round 1
Reviewer 1 Report
This is a nice and well written review dealing with the use of anti-angiogenic molecules in cancer/cancer resistance to classical chemotheraputic drugs. I have a few comments that I would like to be taken into account by the authors.
Introduction : « the initial event…. » the sentence is not clear. Either the « initial event » is a genetic condition (germinal mutation associated to cancer susceptibilty, eg : BRCA1/2) or, dealing with sporadic cancers where it is hard to show in human neoplasia « who » is the first in cancer «initiation ».
detail points :
figure 1. I don’t understand where filipodia are ; they canont be seen in the inset. Their main components are poorly described, eventhough this is not the major topic of the review.
figure 2. to help the reader , the several downstream pathways could be illustrated
5.3 : ROS ; the NFR2, HIF1 scavengers or appropriate DNA repair complexes are not mentioned.
Author Response
Reviewer 1
This is a nice and well written review dealing with the use of anti-angiogenic
molecules in cancer/cancer resistance to classical chemotheraputic drugs. I have
a few comments that I would like to be taken into account by the authors.
- Thank you very much for your comments and criticisms.
Introduction : « the initial event…. » the sentence is not clear. Either the « initial
event » is a genetic condition (germinal mutation associated to cancer
susceptibilty, eg : BRCA1/2) or, dealing with sporadic cancers where it is hard to
show in human neoplasia « who » is the first in cancer «initiation ».
- We agree. This part of the sentence was removed and the relevance of
angiogenesis in cancer progression was maintained.
detail points :
figure 1. I don’t understand where filipodia are ; they canont be seen in the inset.
Their main components are poorly described, eventhough this is not the major
topic of the review.
- We agree with your suggestion and tip cells morphology was altered in order to
present filipodia, whose structure is briefly described in the figure legend.
figure 2. to help the reader, the several downstream pathways could be illustrated
- We agree and the main signaling pathways related to the receptors targeted by
anti-angiogenic drugs were added to figure 2.
5.3 : ROS ; the NFR2, HIF1 scavengers or appropriate DNA repair complexes
are not mentioned.
- A new section (5.3) dedicated to the impact of drugs targeting DNA repair in
cancer angiogenesis, was added. The fact that oxidative stress regulators and
the generation of free radicals are relevant in mutagenesis and in tumor
angiogenesis was reinforced.
Reviewer 2 Report
In the following, I would like to comment on the review paper by Ms. Lopes Coelho. The paper deals with the effect of antiangiogenic therapy on the tumor cell and tumor stroma, its limitations in the past and its chance in the future with the combination of immunotherapies. The presented review is very detailed, well structured and with very successful illustrations.
Subjectively, there is little to improve or suggest for improvement. I would like to see a table with promising ongoing studies on the combination of anti-angiogenesis + immunotherapy. I would like to congratulate the authors for this successful work.
Author Response
Reviewer 2
In the following, I would like to comment on the review paper by Ms. Lopes Coelho.
The paper deals with the effect of antiangiogenic therapy on the tumor cell and
tumor stroma, its limitations in the past and its chance in the future with the
combination of immunotherapies. The presented review is very detailed, well
structured and with very successful illustrations.
Subjectively, there is little to improve or suggest for improvement. I would like to
see a table with promising ongoing studies on the combination of antiangiogenesis + immunotherapy. I would like to congratulate the authors for this
successful work.
- Thank you very much for your comments.
- Table 1 was added to the manuscript, it presents clinical trials on the combined
testing of anti-antiangiogenic therapy and immunotherapy